# The Effect of Heparin and Other Exogenous Glycosaminoglycans (GAGs) in Reducing IL-1β-Induced Pro-Inflammatory Cytokine IL-8 and IL-6 mRNA Expression and the Potential Role for Reducing Inflammation

**DOI:** 10.3390/ph17030371

**Published:** 2024-03-14

**Authors:** Murtaza Jafri, Lin Li, Binhua Liang, Ma Luo

**Affiliations:** 1Faculty of Medicine, University of Manitoba, Winnipeg, MB R3T 2N2, Canada; jafrim@myumanitoba.ca; 2National Microbiology Laboratory, Winnipeg, MB R3E 3R2, Canada; lin.li@phac-aspc.gc.ca (L.L.); ben.liang@phac-aspc.gc.ca (B.L.); 3Department of Biochemistry and Medical Genetics, University of Manitoba, Winnipeg, MB R3T 2N2, Canada; 4Department of Medical Microbiology and Infectious Diseases, University of Manitoba, Winnipeg, MB R3T 2N2, Canada

**Keywords:** heparin, glycosaminoglycans, IL-8, IL-6, IL-1β, inflammation

## Abstract

Glycosaminoglycans (GAGs) are long linear polysaccharides found in every mammalian tissue. Previously thought only to be involved in cellular structure or hydration, GAGs are now known to be involved in cell signaling and protein modulation in cellular adhesion, growth, proliferation, and anti-coagulation. In this study, we showed that GAGs have an inhibitory effect on the IL-1β-stimulated mRNA expression of IL-6 and IL-8. Exogenous heparin (*p* < 0.0001), heparan (*p* < 0.0001), chondroitin (*p* < 0.049), dermatan (*p* < 0.0027), and hyaluronan (*p* < 0.0005) significantly reduced the IL-1β-induced IL-8 mRNA expression in HeLa cells. Exogenous heparin (*p* < 0.0001), heparan (*p* < 0.0001), and dermatan (*p* < 0.0027) also significantly reduced IL-1β-induced IL-6 mRNA expression in HeLa cells, but exogenous chondroitin and hyaluronan had no significant effect. The exogenous GAGs may reduce the transcription of these inflammatory cytokines through binding to TILRR, a co-receptor of IL-1R1, and block/reduce the interactions of TILRR with IL-1R1.

## 1. Introduction

GAGs are linear polysaccharides made of repeating disaccharide units containing an amino sugar derivative, usually glucosamine or galactosamine [1]. Previously thought only to be involved in cellular structure or hydration, GAGs are now known to be involved in cell signaling and protein modulation through their binding activities in cellular adhesion, growth, proliferation, and anti-coagulation [2,3]. GAGs can be categorized into five primary groups based on differences in a core disaccharide [4]: heparin/heparan sulfate, dermatan sulfate, chondroitin sulfate, keratan sulfate, and hyaluronic acid. This study focuses particular interest on five well-known GAGs: heparin, heparan sulfate, dermatan sulfate, chondroitin sulfate, and hyaluronan [5,6,7]. These glycosaminoglycans and their structures are described in Table 1.

Proteoglycans and glycosylated proteins act as co-receptors in a number of pathways, including immune function and inflammation. These molecules interact with the receptor complex to influence receptor function and binding action or are involved with signal activation and disease [8,9,10]. Numerous studies have outlined the role of Toll-like and IL-1 receptors in being critical controllers of immunity and inflammation [11,12,13]. Recently, research has shown that a heparan sulfate proteoglycan [14], TILRR, is a co-receptor of IL-1R1 (type I signaling receptor of the IL-1 receptor subfamily) [15,16]. The binding of TILRR to the IL-1R1 magnifies the NF-kB (nuclear factor kappa-light-chain enhancer of activated B cells) regulated inflammatory response [15,17]. Specifically, it was shown that TILRR over-expression potentiates NF-κB signaling ligands IL-8 and IL-6 through immune and inflammation-induced gene expression [18,19].

IL-8 is a pro-inflammatory chemokine ligand that plays an important role in the induction of innate immune defenses via the recruitment of innate immune cells, such as neutrophils, to the site of inflammation [20,21]. IL-8 production occurs in several cell types, such as fibroblasts, endothelial cells, blood monocytes, and other non-immune cells [22,23,24]. IL-8 expression is insignificant in unstimulated cells; however, when cells are stimulated by other pro-inflammatory cytokines, such as IL-1β, which is an acute and chronic mediator of inflammation, IL-8 expression is induced [22,23,24]. Similar to IL-8, IL-6 is a pro-inflammatory cytokine mainly produced by monocytes and macrophages [25]. IL-6 production is also stimulated by IL-1β [23].

Previous studies have established that NF-κB is activated in a RAS-dependent activation pathway through IL-1 binding to AcP (IL-1 accessory protein) [26] and its co-receptor TILRR [15]. In cardiac fibroblasts, IL-1β-mediated activation of NF-κB contributes to IL-1β-induced IL-8 mRNA expression [24]. TILRR has two GAG binding sites through which TILRR binds to IL-1R1 to enhance IL-1β-NF-κB inflammatory responses [15].

Clinically, it is imperative to control inflammation and immune function to improve patient outcomes after surgery. Cardiac surgery using cardiopulmonary bypass has shown increased serum and myocardium levels of NF-κB and pro-inflammatory cytokines such as IL-6 and IL-8 [22,27,28,29,30,31,32]. In some cases, acute activation of NF-κB has been shown to have a protective role against cell death in cardiac myocytes through repression of apoptosis following ischemic myocardial injury [33,34,35]. However, prolonged activation of NF-κB appears to have maladaptive pathologies following ischemia and reperfusion of the heart through the chronic expression of inflammatory cytokines, such as IL-6, leading to cardiac cell death [36,37]. Up to at least 4 weeks following a coronary ligation procedure, clinical cases have displayed elevated NF-κB, promoting adverse remodeling through pro-inflammatory, pro-fibrotic, and pro-apoptotic effects leading to eventual heart failure [38,39]. The levels of intramyocardial IL-6 mRNA are significantly upregulated, over 50-fold in the infarcted area and 15-fold in the non-infarcted myocardium, leading to the increased adverse remodeling of heart tissue [39]. In addition, serum levels of IL-8 have been shown to increase transiently in patients following acute myocardial infarctions [40,41]. The proposed mechanism of myocardial damage is due to neutrophils infiltrating the infarcted myocardium and the subsequent production of IL-8, leading to the further accumulation and activation of neutrophils, thus increasing the degree of myocardial damage [42,43]. Ultimately, these cytokines are released due to tissue injury from ischemia and could be a cause of reperfusion injuries in bypass surgeries [22]. Therefore, controlling IL-6 and IL-8 mRNA transcription or inhibiting NF-κB-induced inflammation may be an important tool for clinicians in improving patient recovery and outcomes.

The current study aims to evaluate the effect of five different GAGs (heparin, heparan, chondroitin, dermatan, and hyaluronan), on the IL-1β-induced mRNA expression of cytokines IL-6 and IL-8 and evaluate their effect in reducing inflammatory cytokine expression, such as IL-6 and IL-8; and how they may impact the way clinical cases are treated or how clinicians could reduce inflammation via control of IL-6 and IL-8 mRNA transcription using pharmacological agents familiar to them, leading to better patient outcomes. These GAGs were chosen as they are commonly used in healthcare.

## 2. Results

### 2.1. The Effect of GAGs on IL-1β-Stimulated Expression of IL-6 mRNA

The effect of each of the five GAGs on the IL-1β-stimulated production of IL-6 mRNA within HeLa cells was compared with controls without the treatment of GAGs (Figure 1). A significant inhibitory effect on IL-6 mRNA expression induced by IL-1β was observed with the treatment of three of the five GAGs, notably heparin, heparan, and dermatan. As expected, the levels of IL-6 mRNA expression were significantly higher in the controls without these GAGs. Heparin (68% reduction, *p* < 0.0001), heparan (56% reduction, *p* < 0.0001), and dermatan (13% reduction, *p* < 0.0027) treatments of HeLa cells all showed a statistically significant reduction in the IL-1β-induced expression of IL-6 mRNA. In contrast, the chondroitin (*p* < 0.547) and hyaluronan (*p* < 0.335) treatments showed no significant effect on the IL-6 mRNA expression induced by IL-1β, as shown in Figure 1.

### 2.2. The Effect of GAGs on IL-1β-Stimulated Expression of IL-8 mRNA

The effect of each of the five GAGs on the IL-1β-induced expression of IL-8 mRNA in HeLa cells was compared with controls without GAG treatment (Figure 2). After IL-1β stimulation, the levels of IL-8 mRNA were significantly increased in the controls, as expected. However, there was a significant inhibitory effect on the IL-8 mRNA expression induced by IL-1β after treatment with all five GAGs (Figure 2). With the treatment of heparin, heparan, chondroitin, dermatan, and hyaluronan, the HeLa cells expressed significantly lower levels of IL-8 mRNA compared to cells without the GAGs after IL-1β stimulation. Heparin (93% reduction, *p* < 0.0001), heparan (95% reduction, *p* < 0.0001), dermatan (26% reduction, *p* < 0.0027), hyaluronan (27% reduction, *p* < 0.0005), and chondroitin (7% reduction, *p* < 0.049) treatments significantly reduced the expression of IL-8 mRNA induced by IL-1β in HeLa cells, as shown in Figure 2.

## 3. Discussion

In this study, we demonstrated that exogenous GAGs, specifically heparin, heparan, chondroitin, dermatan, and hyaluronan, significantly suppressed IL-1β-stimulated IL-8 mRNA expression in HeLa cells, while heparin, heparan, and dermatan significantly suppressed IL-1β-stimulated IL-6 mRNA expression in HeLa cells. IL-8 is a critical inflammatory mediator that functions as a chemotactic factor for neutrophils and lymphocytes [44,45]. A significant increase in serum IL-6 and IL-8 levels has been reported in patients undergoing heart transplantation and animal studies [38,39,40]. These observations suggested that IL-6 and IL-8 are involved in cardiac inflammation [46]. IL-1β is a most studied member of the IL-1 family and a pro-inflammatory mediator in acute and chronic inflammation, expressed under various disease conditions, such as ischemia, periodontitis, and bone loss [47]. Controlling IL-1β-mediated inflammation is important because it has been shown that the blockade of IL-1 signaling reduces damage from heart failure following acute myocardial infarction in humans [48]. The observations suggest that IL-1β, IL-6, and IL-8 play an important role in myocardial infarction pathogenesis and other inflammatory diseases and may contribute to inflammation-resolved damage. We showed that in this study, exogenous GAGs, heparin, heparan, chondroitin, dermatan, and hyaluronan can suppress IL-1β-induced IL-6 and IL-8 mRNA expression in HeLa cells. The result supports future clinical evaluation of using heparin and other GAGs in reducing patients’ serum levels of IL-6 and IL-8 in a hospital setting.

Based on available publications, the exogenous GAGs may suppress IL-1β-induced IL-6 and IL-8 mRNA expression through three possible ways: (1) Exogenous GAGs can induce the expression of IL-1Ra that competes with IL-1β for IL-1R, thus blocking the IL-1 receptor inflammation signal-transduction pathway [49,50,51]. (2) Exogenous GAGs bind to IL-1β [52], thus reducing the amount of IL-1β in the treatment; studies showed that GAGs selectively interact with chemokines [53], and the bindings of IL-1β to different GAGs is variable with heparin, which has the highest binding to IL-1β, and chondroitin, which has the lowest binding to IL-1β [52]. Thus, binding of GAGs to IL-1β and reducing the available IL-1β for IL-R1 is a possibility. (3) Exogenous GAGs compete with TILRR for IL-1R1, thus reducing/blocking the binding of TILRR to IL-1R1 [14]. The proposed possible mechanism is shown in Figure 3.

## 4. Materials and Methods

### 4.1. HeLa Cells and Culture Preparation

HeLa cells were chosen as they are human cell lines and were previously screened to have a number of receptor systems, including IL-1R1, and were evaluated for the effect of TILRR on IL-6 and IL-8 (15). The human (female) malignant cervical epithelial carcinoma cell line, HeLa, was provided by the National Institute of Health AIDS reagent program (Catalog #ARP-153, Bethesda, MD, USA) and was maintained in Dulbecco’s Modified Eagle’s Medium (DMEM) (Catalog# D5796, lot# RNB08228, MilliporeSigma Canada Ltd., Oakville, ON, Canada) supplemented with 10% FBS (fetal bovine serum; Catalog# F1051-500ml, lot# 15A407, MilliporeSigma Canada Ltd., Oakville, ON, Canada), 1% glutamin, and 1% antibiotic–antimycotic (Gibco, Catalog# 15240-062, lot# 1030612, Fisher Scientific Company, Ottawa, ON, Canada).

To study the effect of exogenous GAGs on IL-1β-induced IL-6 and IL-8 mRNA expression, we used the cervical HeLa cells plated on the fibronectin-coated 24-well plates. The HeLa cells were stimulated with IL-1β in the presence of one of five GAGs (heparin, heparan, dermatan, hyaluronan, and chondroitin) or a GAG vehicle control (nuclease-free water), and then the cell lysate was collected at 0 h and 24 h.

### 4.2. Fibronectin

Fibronectin (Catalog# F2006-2MG, lot# SLBK4880V, MilliporeSigma Canada Ltd., Oakville, ON, Canada) was dissolved in 2 mL of nuclease-free water for a 1 mg/mL concentration and topped up to 200 mL with phosphate-buffered saline (PBS) pH 7.2 to a final concentration of 10 µg/mL and filter sterilized with a 0.2 µM filter (Corning, Catalog#, 430773, Fisher Scientific, Ottawa, ON, Canada). The fibronectin was used to coat 24-well tissue culture plates.

### 4.3. IL-1β

IL-1β (Catalog# I9401-5µg, lot#SLBB897V, MilliporeSigma Canada Ltd., Oakville, ON, Canada) was dissolved in 50 µL nuclease-free water, topped up to 5 mL with PBS pH 7.2. The final concentration was 57.8 nM. This was diluted to 1 nM for the desired treatment concentration.

### 4.4. The Preparation of GAGs

Heparin: heparin sodium salt (Catalog#H3393-10Kµ, lot# SLBL1083V, MilliporeSigma Canada Ltd., Oakville, ON, Canada) was dissolved in nuclease-free water for a final concentration of 10 µg/mL and filter sterilized with a 0.2 µM filter (Corning, Catalog#, 431219, lot #09301015, Fisher Scientific, Ottawa, On, Canada). Heparan: heparan sulfate sodium salt (Catalog# H7640, lot# SLBN5533V, MilliporeSigma Canada Ltd., Oakville, ON, Canada) was dissolved in nuclease-free water for a final concentration of 1 mg/mL and filter sterilized with a 0.2 µM filter. Chondroitin: chondroitin sulfate salt from shark cartilage (C4384-250mg, lot# 1426300V, MilliporeSigma Canada Ltd., Oakville, ON, Canada) was dissolved in nuclease-free water for a final concentration of 50 mg/mL and filter sterilized with a 0.2 µM filter. Dermatan: dermatan sulfate sodium salt (Ltd., Oakville, ON, Canada. Catalog# 1171455-25MG, MilliporeSigma Canada Ltd., Oakville, ON, Canada) was dissolved in nuclease-free water for a final concentration of 1 mg/mL and filter sterilized with a 0.2 µM filter. Hyaluronan: hyaluronate sodium (Lifecore Biomedical, Catalog# HA5K-1, Cedarlane, Burlington, ON, Canada) was dissolved in nuclease-free water for a final concentration of 1 mg/mL and filter sterilized with a 0.2 µM filter. The treatment concentration of all GAGs was 1 × 10^−5^ M per well.

### 4.5. Experimental Procedures

The first three columns of 24-well culture plates (half of the plate) (Costar, Catalog#3524, Fisher Scientific, Ottawa, ON, Canada) were coated with fibronectin (221 µL of fibronectin at 10 µg/mL was used to coat each well), and the other half of the plate served as a no-fibronectin control. The plates were dried in a biological safety cabinet (BSC) for 1 h, then aspirated, sealed, and stored at 4 °C. HeLa cells were washed 2× with 40 mL of serum-free DMEM and diluted to a concentration of 1.25 × 10^5^ cells/mL. Cells were ~50% confluent with normal morphology. In the “GAG” treatment wells, 50 µL of the specific GAG was added, and the controls had 50 µL of nuclease-free water added. Plates were incubated overnight. A total of 50 µL of IL-1β was then added to each of the wells to start treatment (0 h time point).

Cells and supernatants were harvested at the 0 h time point and 24 h time point. The cell supernatant was transferred to a 1.5 mL LoBind micro tube and spun at 10,000× *g* for 10 min at 4 °C. The supernatant was aliquot into three LoBind tubes with 99 µL in each tube, and 11 µL of 5% bovine serum albumin (BSA) (final concentration 0.5%) was added to each of the microcentrifuge tubes. The HeLa cells were harvested by adding 350 µL of buffer RLT from the RNeasy mini kit (Qiagen, Markham, ON, Canada. Catalog# 74104, lot#433163498) to each well, rinsed 5×, and then collected in a 1.5 mL tube. All collected samples were stored in a −80 °C freezer. The RNA was isolated with the RNeasy Mini kit (Qiagen, Markham, ON, Canada. Catalog#74104, lot#433163498) and QIAshredder for disruption and homogenization (Qiagen, Markham, ON, Canada. Catalog#79654, lot# 139309413). This was performed on column DNase digestion (Qiagen, Markham, ON, Canada. Catalog# 79254, lot#151017865). Samples were then reverse transcribed with the RT2 first strand kit (Qiagen, Markham, ON, Canada. Catalog# 330404, lot#100508155), 400 ng per sample, and topped to 8 µL with nuclease-free water in the gDNA elimination step.

### 4.6. Real-Time Quantitative Reverse-Transcription PCR (qRT-PCR)

RNA samples were evaluated via RT2 RNA QC PCR array (Qiagen, Markham, ON, Canada. Catalog# PAHS-9992A-1, 330291, lot #745112) with RT2 SYBR(R) Green ROX qPCR Mastermix (Qiagen, Markham, ON, Canada, Catalog# 330523). We used 1 μL of cDNA in a 25 μL reaction volume. Amplification of copy DNA (cDNA) was performed in 40 cycles, consisting of a single initial cycle at 95 °C for 10 min, followed by 40 cycles run at 95 °C for 15 s, followed by 60 °C for 1 min. The software can automatically calculate the delta–delta Ct for the relative quantification of gene expression. Data were exported and organized in a Microsoft Office Excel sheet and analyzed with GeneGlobe Data Analysis Centre (Qiagen, Markham, ON, Canada). An applied BioSystem 7900 HT Fast Real-time PCR 96-well standard block (Fisher Scientific, Ottawa, ON, Canada) was used for all qRT-PCR analysis. A Student t-test with 95% CI was performed for the statistical analysis using GraphPad Prism version 9.4 (GraphPad, Boston, MA, USA). All *p* < 0.05 were reported and indicated using asterisks * *p* < 0.05, ** *p* < 0.01, *** *p* < 0.001, and **** *p* < 0.0001.

In addition, real-time quantification for the two targeted immune-responsive genes, IL-8 (Qiagen, Markham, ON, Canada. Catalog# PPH00568A-200) and IL-6 (Qiagen, Markham, ON, Canada. Catalog# PPH00560C-200), and three reference genes (Qiagen, Markham, ON, Canada. Catalog# PPH01018C-200), (Qiagen, Markham, ON, Canada. Catalog# PPH21138F-200), and (Qiagen, Markham, ON, Canada. Catalog# PPH01094E-200), was used to measure the mRNA transcript expression.

## 5. Conclusions

GAGs, such as heparin, have been used clinically in hospitals for years for surgery or other hospital needs [56,57]. In this study, we demonstrated that GAGs have the ability to inhibit IL-1β-induced expression of IL-6 and IL-8 mRNA in HeLa cells. Although the exact mechanism remains to be further studied, we have seen in previous studies that these pro-inflammatory chemokines can be activated through the overexpression of TILRR [18]. Further studies should be performed to assess the clinical use of GAGs in patients requiring a reduction in the inflammatory response.

## Figures and Tables

**Figure 1 pharmaceuticals-17-00371-f001:**
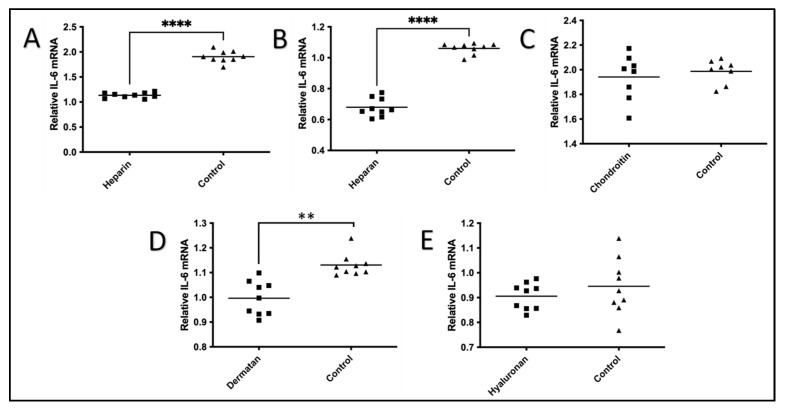
Effects of five GAGs on IL-1β-Induced production of IL-6 in HeLa cells in nine biological replicates. HeLa cells were plated on fibronectin-coated plates and stimulated with IL-1β (10 ng/mL) for 24 h with or without one of the five GAGs. The HeLa cells were collected, and the RNA was isolated. The levels of IL-6 mRNA in each GAG treatment were measured using QRT-PCR at 0 h and 24 h using the 2−∆∆Ct method. (**A**) Heparin; (**B**) heparan; (**C**) chondroitin; (**D**) dermatan; and (**E**) hyaluronan. Student *t*-test with 95% CI was performed for the statistical analysis using GraphPad Prism version 9.4. All *p* < 0.05 were reported and indicated using asterisks ** *p* < 0.01, and **** *p* < 0.0001. The values of the GAGs were plotted in a square symbol, while the values of the controls were plotted in a triangle symbol.

**Figure 2 pharmaceuticals-17-00371-f002:**
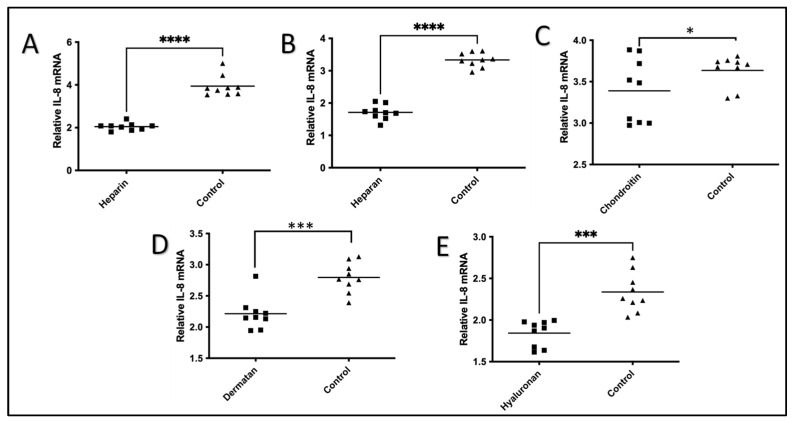
Effects of five GAGs on IL-1β-induced production of IL-8 in HeLa Cells in nine biological replications. HeLa cells were plated on fibronectin-coated plates and stimulated with IL-1β (10 ng/mL) for 24 h with or without one of the five GAGs. The HeLa cells were collected, and the RNA was isolated. The levels of IL-8 mRNA in each GAG treatment were measured using QRT-PCR at 0 h and 24 h using the 2−∆∆Ct method. (**A**) Heparin; (**B**) heparan; (**C**) chondroitin; (**D**) dermatan; and (**E**) hyaluronan. A Student *t*-test with 95% CI was performed for the statistical analysis using GraphPad Prism version 9.4. All *p* < 0.05 were reported and indicated using asterisks * *p* < 0.05, *** *p* < 0.001, and **** *p* < 0.0001. The values of the GAGs were plotted in a square symbol, while the values of the controls were plotted in a triangle symbol.

**Figure 3 pharmaceuticals-17-00371-f003:**
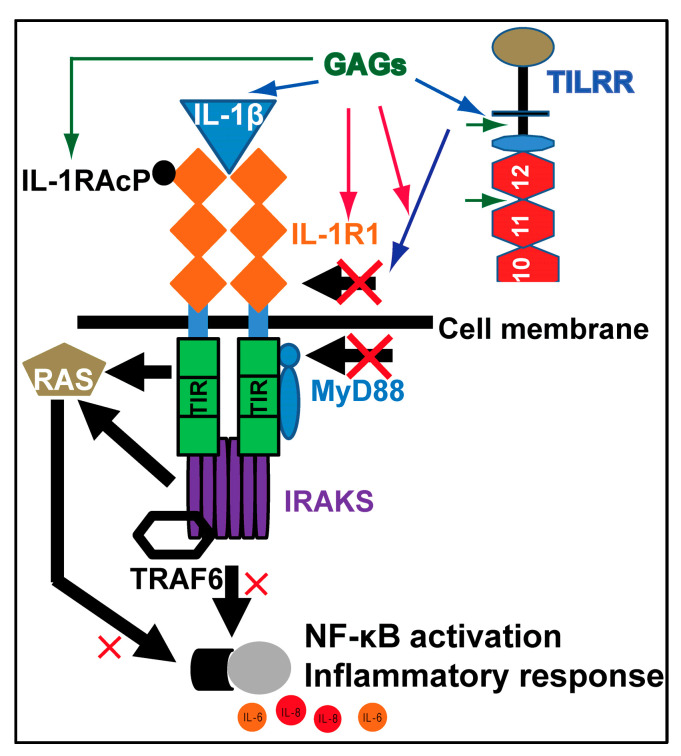
The possible mechanism of exogenous GAGs on reducing IL-1β-induced expression of IL-6 and IL-8 mRNA. TILRR has two GAG binding sites (the locations are indicated by the two small green arrows). TILRR associates with the extracellular domain of IL-1R1 [15] and functions as a co-receptor of IL-1R1. TILRR is not only expressed in cells and tissues but also circulates in blood [54,55]. The associations of heparin or other GAGs with TILRR could block/reduce TILRR’s interaction with IL-1/IL-R1, thus reducing the binding of MyD88 to TIR and leading to the reduced expression of inflammation-responsive genes and lower NF-κB activation and inflammatory cytokine expression, including IL-6 and IL-8 [18]. The blue arrow and black arrow above the cell membrane indicate the association of TILRR with the IL-R1 receptor complex. The blue arrow also indicates GAGs’ association with IL-1β and TILRR. The red arrows indicate that exogenous GAGs may interfere with the association of TILRR with the IL-R1 receptor complex through its binding to TILRR via the two GAG attachment sites indicated by the two small green arrows. The green arrow from GAGs to IL-RA indicates positive regulation of GAGs on IL-RA expression.

**Table 1 pharmaceuticals-17-00371-t001:** Glycosaminoglycans and structures [1].

Glycosaminoglycan (GAG)	Disaccharide Repeat
Heparin	α4GlcNAc-β4GlcUA
Heparan sulfate	α4GlcNAc-β4GlcUA(SO_4_)
Dermatan sulfate	β3GalNAc-β4GlcUA(SO_4_)
Chondroitin sulfate	β3GalNAc-β4GlcUA(SO_4_)
Hyaluronan	β3GlcNAc-β4GlcUA

Notes: a. GlcNAc = N-acetylglucosamine, GlcUA = glucuronic acid, GalNAc = N-Acetylgalactosamine. b. Several modifications are found in most of these GAGs (including O-sulphation of hydroxyls, deacetylation, and N-sulphation or epimerization of glucuronic acid (GlcUA) to iduronic acid (IdoUA)).

## Data Availability

All data are contained within the article, and accession numbers are provided in respective sections.

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
