# Peer review of "The Effect of Heparin and Other Exogenous Glycosaminoglycans (GAGs) in Reducing IL-1β-Induced Pro-Inflammatory Cytokine IL-8 and IL-6 mRNA Expression and the Potential Role for Reducing Inflammation"

_pharmaceuticals, 2024, doi:10.3390/ph17030371_

Round 1
Reviewer 1 Report
Comments and Suggestions for Authors
The brief report by Murtafa Jafri et al presents in vitro data on the effect of 5 GAGs on the production of mRNA of 2 pro-inflammatory cytokines, IL-6 and IL-8, by a cell line. Authors want to highlight potential anti-inflammatory beneficial effect of GAG use in surgery.
Have the authors measured IL-6 and IL-8 release into supernatants?
Methods p6, line 197-217: restructure to avoid repetition
The GAG preparation can be presented in one paragraph with the general preparation then the list of individual specifications for each GAG.
Figure 1 and Figure 2 show 5 different graphs plotting the same results: relative expression of mRNA, please pull the 5 GAG in one plot (with or without vertical lines to separate each GAG). GAG’s names can be placed on x axis
As GAG and control symbols are different (squares and triangles), control does not have to be repeated 5 times, and symbols should be explained in legend
title: change production by transcription, as production implies that the cytokine was measured in supernatants
Figure 3: not clear enough
The title of the figure should give a good idea of what the figure is about
IL1b is named IL1, exogenous GAG is named Heparin, IL-6 and IL-8 mRNA are not shown.
The association between IL-1R1 and TILRR is not obvious at all (is it black arrow?)
Why is TILRR shown above the cell membrane.
The 2 red arrows below heparine and blue arrow going down do not sense easily and are not explained.
Minor:
line 61: IL-6 is not a chemokine (as IL-8). Please rephrase
line 223: use DMEM abbreviation previously cited
Author Response
A point-by-point response to the reviewers’ comments:
Reviewer 1: Comments and Suggestions for Authors
The brief report by Murtafa Jafri et al presents in vitro data on the effect of 5 GAGs on the production of mRNA of 2 pro-inflammatory cytokines, IL-6 and IL-8, by a cell line. Authors want to highlight potential anti-inflammatory beneficial effect of GAG use in surgery.
Have the authors measured IL-6 and IL-8 release into supernatants?
Our response: We thank the reviewer’s comment. In this study we have not measured the IL-6 and IL-8 release into supernatants. Although measuring mRNA transcription may not exactly represent the amount of protein released into supernatants, it does show the effect of treatment of GAGs is on IL-1β induced Il-6 and IL-8 gene transcription. Studies showed that GAGs can bind to multiple cytokines/chemokines, including IL-8 and IL-6 (1. Kuschert GS, Coulin F, Power CA, Proudfoot AE, Hubbard RE, Hoogewerf AJ, Wells TN: Glycosaminoglycans interact selectively with chemokines and modulate receptor binding and cellular responses. Biochemistry (Mosc) 38: 12959–12968, 1999. 2. Xie X, Rivier AS, Zakrzewicz A, Bernimoulin M, Zeng XL, Wessel HP, Schapira M, Spertini O: Inhibition of selectin-mediated cell adhesion and prevention of acute inflammation by nonanticoagulant sulfated saccharides. J Biol Chem 275:34818–34825, 2000. 3. Ramsden L, Rider CC. Selective and differential binding of interleukin (IL)-1 alpha, IL-1 beta, IL-2 and IL-6 to glycosaminoglycans. Eur J Immunol 1992 Nov;22(11):3027-31. doi: 10.1002/eji.1830221139.), the GAGs added in the experiment may influence quantification of the released IL-6 and IL-8 protein in the study. Thus, we can only conclude that “The Effect of Heparin and Other Exogenous Glycosaminoglycans (GAGs) In Reducing IL-1β Induced Pro-inflammatory Cy-tokines IL-8 and IL-6 mRNA Expression and Potential Role for Reducing Inflammation” for this brief report.
Methods p6, line 197-217: restructure to avoid repetition The GAG preparation can be presented in one paragraph with the general preparation then the list of individual specifications for each GAG.
Our response: We revised the method section according to the reviewer’s comment.
Figure 1 and Figure 2 show 5 different graphs plotting the same results: relative expression of mRNA, please pull the 5 GAG in one plot (with or without vertical lines to separate each GAG). GAG’s names can be placed on x axis
Our response: We thank the reviewer for the comment. However, although the controls for all five GAGs are treated with nuclease-free water, the experiments were conducted separately and are biological repeat. As with all biological experimental repetitions their values are variable. So, the data be best presented as a complex figure containing five individual figures representing the data for each of the GAGs.
As GAG and control symbols are different (squares and triangles), control does not have to be repeated 5 times, and symbols should be explained in legend.
Our response: In response to the reviewer’s comments we have added explanations of the symbols in the Figure legends.
title: change production by transcription, as production implies that the cytokine was measured in supernatants
Our response: The reviewer may mean the change should be made in the abstract. We revised this in the abstract.
Figure 3: not clear enough. The title of the figure should give a good idea of what the figure is about. IL1b is named IL1, exogenous GAG is named Heparin, IL-6 and IL-8 mRNA are not shown. The association between IL-1R1 and TILRR is not obvious at all (is it black arrow?) Why is TILRR shown above the cell membrane. The 2 red arrows below heparine and blue arrow going down do not sense easily and are not explained.
Our response: In response to the reviewer’s comments we revised Figure 3. We also revised the Figure legend and added related references to better explain the figure. The revision is indicated by the red colored fond.
Minor:
line 61: IL-6 is not a chemokine (as IL-8). Please rephrase
Our response: Thank you. We corrected it in the revision.
line 223: use DMEM abbreviation previously cited
Our response: Revised as suggested by the reviewer.

Reviewer 2 Report
Comments and Suggestions for Authors
The Effect of Heparin and Other Exogenous Glycosaminoglycans (GAGs) In Reducing IL-1β Induced Pro-inflammatory Cytokines IL-8 and IL-6 mRNA Expression and Potential Role for Reducing Inflammation
In the present manuscript, the effect of heparin, heparan, chrondroitin, dermatan, and hyaluronan was investigated on the IL-1β stimulated mRNA expression induced IL-6 and IL-8 in HeLa cells. A significant inhibitory effect on IL-6 mRNA expression induced by IL-1β was observed with treatment heparin, heparan, and dermatan. All five glycosaminoglycans inhibited IL-8 mRNA expression induced by IL-1β. The authors propose two mechanisms for explaining the results.
The manuscript is well written, the data nicely presented, however I feel that the discussion needs to be improved. Heparin is also known to bind various chemokines such as IL-8 (Kurschert et al., 1999). Heparin-chemokine complexes were unable to bind to the receptor, resulting in a block of the biological activity (Xie et al., 2000). Thus, the logical explanation is that those glycosaminoglycans are binding to IL-1b. This should be addressed not only in the discussion, but also experimentally.
Kuschert GS, Coulin F, Power CA, Proudfoot AE, Hubbard RE, Hoogewerf AJ, Wells TN: Glycosaminoglycans interact selectively with chemokines and modulate receptor binding and cellular responses. Biochemistry (Mosc) 38: 12959–12968, 1999.
Xie X, Rivier AS, Zakrzewicz A, Bernimoulin M, Zeng XL, Wessel HP, Schapira M, Spertini O: Inhibition of selectin-mediated cell adhesion and prevention of acute inflammation by nonanticoagulant sulfated saccharides. J Biol Chem 275:34818–34825, 2000.
Author Response
A point-by-point response to the reviewers’ comments:
Reviewer 2: Comments and Suggestions for Authors
The Effect of Heparin and Other Exogenous Glycosaminoglycans (GAGs) In Reducing IL-1β Induced Pro-inflammatory Cytokines IL-8 and IL-6 mRNA Expression and Potential Role for Reducing Inflammation
In the present manuscript, the effect of heparin, heparan, chrondroitin, dermatan, and hyaluronan was investigated on the IL-1β stimulated mRNA expression induced IL-6 and IL-8 in HeLa cells. A significant inhibitory effect on IL-6 mRNA expression induced by IL-1β was observed with treatment heparin, heparan, and dermatan. All five glycosaminoglycans inhibited IL-8 mRNA expression induced by IL-1β. The authors propose two mechanisms for explaining the results.
The manuscript is well written, the data nicely presented, however I feel that the discussion needs to be improved. Heparin is also known to bind various chemokines such as IL-8 (Kurschert et al., 1999). Heparin-chemokine complexes were unable to bind to the receptor, resulting in a block of the biological activity (Xie et al., 2000). Thus, the logical explanation is that those glycosaminoglycans are binding to IL-1b. This should be addressed not only in the discussion, but also experimentally.
Our response: We thank the reviewer for the very valuable comments. In response to the reviewer’s comments we searched for publications related to the IL-1β binding to GAGs and find the publication by Lawrence Ramsden and Christopher Rider (Eur. J. Immunol. 1992 22:3027-3031). The published study evaluated the binding of several cytokines, including IL-1β to several GAGs, and provide important information. We added this information in the discussion of the revised manuscript.

Round 2
Reviewer 2 Report
Comments and Suggestions for Authors
Its ok.